# 'Overnight, things changed. Suddenly, we were in it': a qualitative study exploring how surgical teams mitigated risks of COVID-19

Daisy Elliott [1], Cynthia Ochieng,[1] Marcus Jepson,[2] Natalie S Blencowe [1,3] Kerry NL Avery [1], Sangeetha Paramasivan,[2] Sian Cousins,[1] Anni Skilton,[1] Peter Hutchinson [4], David Jayne,[5] Martin Birchall,[6] Jane M Blazeby [1,3] Jenny L Donovan,[2] Leila Rooshenas[2]

For numbered affiliations see end of article.

**Correspondence to**
Dr Daisy Elliott;
daisy.elliott@bristol.ac.uk

## ABSTRACT

**Objectives** COVID-19 presents a risk of infection and transmission for operating theatre teams. Guidelines to protect patients and staff emerged and changed rapidly based on expert opinion and limited evidence. This paper presents the experiences and innovations developed by international surgical teams during the early stages of the pandemic to attempt to mitigate risk.

**Design** In-depth, semistructured interviews were audio recorded, transcribed and analysed thematically using methods of constant comparison.

**Participants** 43 participants, including surgeons from a range of specialties (primarily general surgery, otolaryngology, neurosurgery, cardiothoracic and ophthalmology), anaesthetists and those in nursing roles.

**Setting** The UK, Italy, Spain, the USA, China and New Zealand between March and May 2020.

**Results** Surgical teams sought to mitigate COVID-19 risks by modifying their current practice with an abundance of strategies and innovations. Communication and teamwork played an integral role in how teams adapted, although participants reflected on the challenges of having to improvise in real time. Uncertainties remained about optimal surgical practice and there were significant tensions where teams were forced to balance what was best for patients while contemplating their own safety.

**Conclusions** The perceptions of risks during a pandemic such as COVID-19 can be complex and context dependent. Management of these risks in surgery must be driven by evidence-based practice resulting from a pragmatic and novel approach to collation of global evidence. The context of surgery has changed dramatically, and surgical teams have developed a plethora of innovations. There is an urgent need for high-quality evidence to inform surgical practice that optimises the safety of both patients and healthcare professionals as the COVID-19 pandemic unfolds.

## INTRODUCTION

The WHO declared the COVID-19 outbreak a pandemic on 11 March 2020.[1] Many countries have since struggled with escalating case

## Strengths and limitations of this study

► Undertaking qualitative interviews provided important and rich insights into surgical teams' experiences of current practices in the UK and internationally.

► Purposeful sampling ensured that a range of participants were recruited in relation to role, specialty and geographical location.

► Participants were predominately identified via snowball sampling, making it possible that the findings may not be representative of all healthcare professionals involved in the management and treatment of patients undergoing surgical procedures.

► Interviews were conducted during the first wave of the pandemic, so further research is warranted to continue to explore how teams continue to adapt.

numbers and strained healthcare systems.[2] Implementation of infection prevention and control is crucial to deliver healthcare, especially for the personal protection of healthcare workers.[3 4] Operating theatres are particularly vulnerable areas due to high-risk transmission activities such as airway management transmission, aerosol-generating procedures (AGPs) and the involvement of multiple staff.[5] The additional strain presented by a high prevalence of disease, limited resources and staff under pressure greatly increase the risks of transmission and the burden on healthcare systems.[6] Guidelines were rapidly published[7 8] including recommendations from The Royal Colleges (such as the Royal College of Surgeons of England, Royal College of Anaesthetists and The Intensive Care Society), Public Health England and the Department of Health, the American College of Surgeons and the WHO. These have been produced based on expert opinion, surveys, consensus work and rapid reviews, continue

to evolve as evidence unfolds and can include contradictory recommendations.[9–13]

The pandemic inevitably provoked creative solutions to mitigate the risk of infection transmission.[14] Innovation could involve: modifications to decisions around whether and how to operate, surgical techniques, tools and technology, surgical team composition and operating conditions.[15] Although a handful of studies have suggested practical strategies to mitigate COVID-19 risks,[2 9 16–19] qualitative inquiry is a valuable tool for capturing dynamic and complex responses to a pandemic.[20] These methods allow us to understand the ways people make sense of what is happening around them.[21]

The study aimed to explore surgical teams' experiences of current practices during the pandemic. Specific objectives were to (1) understand healthcare professionals' perceived risks of COVID-19 in surgery and (2) to explore how surgical teams mitigated these risks.

## METHODS

Semistructured interviews were conducted with surgical teams in the UK and internationally. This qualitative study adopted principles and techniques described by Glaser and Strauss[22] which enabled the inductive identification of codes from the data to generate findings that were grounded in the data[23] and the constant comparison approach, where new findings are systematically compared with existing data so that similarities and differences can be identified and emerging theories refined through the ongoing assimilation of data.[22 24] An ethics amendment enabled data collection to be conducted via university-approved video conferencing software and for interviews to be conducted internationally. Standards for Reporting Qualitative Research guidelines[25] were followed (online supplemental file 1).

### Patient and public involvement

Public and patient involvement was not conducted as part of the current study.

### Recruitment and sampling

Healthcare professionals involved in the management and treatment of patients undergoing surgical procedures were eligible. A key informant sampling approach was initially adopted,[26] whereby individuals from different surgical specialties who were known to the study team were approached. Subsequent participants were identified via snowball sampling,[26] whereby interviewees recommended potential participants. Participants were also identified through convenience sampling,[27] whereby study details were circulated to members of surgical groups (the National Institute for Health Research Surgical MedTech Co-operative and the Royal College of Surgeons of England). The adverts included instruction to contact the study team if individuals were interested in study participation. As data collection continued, sampling became increasingly purposeful with a view

to achieving a sample of maximum variation, to ensure insights were captured from a range of informants operating in different contexts.[27] We aimed to capture variation in relation to role (including surgeons, theatre nurses and anaesthetists), specialty, years in profession, gender and geographical location. Capturing an international perspective was important to identify how best practice evolved, particularly in those areas/nations that were 'ahead of the curve'. A database of participants and their characteristics was maintained and assessed as the study progressed, and interviews with those who were under-represented were prioritised.

### Data collection

Participants provided written consent to take part in the study. Interviews were conducted either on university-approved secure video conferencing software orvia telephone interviews. Interviews were audio recorded using an encrypted audio recorder or videoing conference software. Interviews were conducted by experienced qualitative researchers (DE, MJ, LR, CO, KA, SP, JD). See online supplemental file 2 for further information on the qualitative researchers' backgrounds.

Semistructured interviews were directed by a topic guide to ensure that the same core areas were consistently covered among the team of interviewers, while allowing flexibility to pursue the detail that was salient to each participant (online supplemental file 3). Specific questions were open ended to encourage the participant to talk about their own experiences of particular topics ('In your own words, can you talk me through how personal protective equipment (PPE) has been used during COVID-19?'), experiences ('Can you describe a case where COVID-19 affected what happened in the operating theatre?') and views ('In your opinion, what are the key risks in surgery due to COVID-19?'). Interviewers were then able to incorporate the interviewee's own terms and concepts into subsequent questions to follow up on specific issues raised and check the interviewer had understood correctly.[28] At the end of the interview, participants were given an opportunity to raise relevant issues that had not already been covered.[29] Regular team meetings allowed for the team to review the topic guide in light of findings and consider potential changes (eg, addition of topics or rephrasing of questions).

Reflexivity is a fundamental part of ensuring the transparency and quality of qualitative research.[30] Interviewers' reflexive notes, which took into account their observations, thoughts and ideas,[31] were shared with the analysis team. Regular team meetings also enabled the interviewers to reflect on the novel process of conducting virtual interviews (a form of functional reflexivity, whereby researchers give critical attention to the way processes influenced research[32]).

### Data analysis

Interviews were transcribed verbatim, checked for accuracy and de-identified. Transcripts were imported into

NVivo (QSR International, USA), where the contents of the transcripts were iteratively coded by three coders (MJ, LR and CO). Codes were identified inductively using methods derived from grounded theory methodology,[22][33] so that findings were grounded in the data to capture and represent healthcare professionals' interpretations and experiences of surgical practice during a pandemic. Through the constant comparison technique, the codes were categorised into themes which were continually compared with the data.[34] In this way, similarities and differences can be identified and emerging themes refined through the ongoing assimilation of data.

A premise of grounded theory is that these comparisons enable identification of 'negative cases' (ie, participants whose perspectives/experiences differed from the main body of evidence).[35][36] This was felt to be important for enhancing the credibility of the analysis as it helped to ensure the findings represented as full and comprehensive an account of participants' experiences and perspectives as possible, rather than those views that were dominant or fit with a particular impression of the results.[22] Data collection continued until no new key themes were identified (data saturation) as determined by the study team.

The coders met regularly to compare their coding and agree on the main broad categories encapsulating all codes. A subset of the transcripts, batched according to coder, were independently double coded by DE to determine broader consistency in coding approaches.[37] Descriptive reports were written on key themes ('Guidance', 'Risk', 'In-theatre processes', 'Organisational', 'Service provision', 'PPE', 'Testing' and 'Physical and mental impact'). This paper focuses on the findings relating to 'Risks', although themes and subthemes were often intertwined and producing descriptive reports provided an opportunity to demonstrate where there was overlap in key findings. Two coders (CO, MJ) also produced a summary on one theme, to explore any major differences in interpreting codes. As multiple qualitative researchers conducted the interviews, one researcher (DE) read all transcripts and synthesised the findings to ensure consistency in analysis and to enable strategic oversight of the findings.[37] Results, with reference to the raw data, were discussed with members of the wider study team (including academic surgeons) and several research participants who were not part of the study team.

## RESULTS

Of the clinicians contacted by the team, nine individuals did not reply to the study invite, said that they were not on the frontline or were unable to find the time for an interview. The final sample of 43 participants included 34 surgeons, 5 anaesthetists and 4 individuals in nursing roles (a practice educator, nurse manager, matron and charge nurse). Surgical specialties included general surgery (n=14), otolaryngology (n=11), neurosurgery (n=5), cardiothoracic surgery (n=2) and ophthalmology (n=1). Thirty-one participants were from the UK, and the remaining 12 participants were from Italy (n=3), China (n=1), Spain (n=3), the USA (n=2) and New Zealand (n=3). Participants were mostly men (77%). Interviews were conducted between March and May 2020 and lasted an average of 47 min (range=21–116 min).

Five subthemes relating to risk were identified: 'Facing new and uncertain risks', 'Changes to the context of surgery due to COVID-19 risk, 'Innovations within high-risk specialties', 'AGPs: complex and uncertain risks' and 'Adapting to PPE-related challenges'. Results are supported with detailed quotations to support the interpretation of data.[30][31] Quotes that were deemed to best illustrate the themes were selected, with careful attention to showcasing different perspectives and negative cases, where relevant.

### Facing new and uncertain risks

All informants described how COVID-19 had had an impact on surgery. At the time of interviews, most elective procedures had been cancelled or postponed and only emergency procedures were being performed:

> We are doing absolute emergency … life and limb only… Even to the extent that some appendicectomy operations are being managed conservatively. So they're just given antibiotics and watch and wait. We know that a lot of people with appendicitis … if they're managed conservatively actually do fine. And a lot of our appendixes now are being managed conservatively and watched. I think if you'd suggested that a year ago, people would have said it was insane. (HP30)

Participants articulated anxiety about contracting the virus. One individual had personally lost a colleague to COVID-19. Many others described how news of clinical professional fatalities had highlighted the sense of personal risk they and their colleagues were being subjected to.

> It is just the shock of my life; I had no idea this was coming. I had no idea that walking into the hospital would be dangerous for me… A huge change in my mindset required to deal with this, because it's changed, almost like a war. Like it is normal one day and then within a week, your normal working life is sort of totally changed. Overnight, things changed… it was in the news, we knew it would happen, but then all of a suddenly, we were in it. (HP7)

While they were accustomed to considering risks for patients, interviewees reflected on the shock of how they suddenly faced new risks to their own health. The risk of transmission to family members added a unique dimension to the risks associated with their roles as healthcare professionals. This was described across a range of surgical specialties and roles.

> And the commonest thing about this whole thing, amongst workers, doctors, nurses, everyone: actually,

everyone's worried about catching it. Everyone is terrified of giving it to their family […] I don't want to get it. I really, really don't want to get it. But I'm infinitely more scared about giving it to my wife or my children. And if I catch it, they'll get it. Almost certainly. Because I'll be infected for several days before I'm symptomatic […] We're not used to processing the risk to us… Because these are unfamiliar risks in our practice… The risks that we've always worried about in the past were risks to our patients, or risks to our career. Rather than risks to our health. And as I say, more particularly risks to the health of our families. (HP4)

## Changes to the context of surgery due to COVID-19 risk

COVID-19 had meant that surgical teams had to rapidly assess all aspects of the patient pathway to minimise the potential spread of the virus. For most, the scale of organisational changes was immense and included changes to hospitals, pathways, staff rotas and operating theatres. There was an urgent need to expand the capacity for intensive care, with comprehensive changes that often meant repurposing operating theatres. There were also considerable discussions about changes to accommodate COVID-19 and non-COVID-19 areas, including separate operating theatres for COVID-19-infected patients and those not infected. These changes required new thinking about patient pathways and processes in operating theatres. Considerable effort went into minimising contact points and the number of people in the operating theatre to avoid potential infection transmission and contamination.

It is a massive move. Everyone will look at the pathway for every scenario. We are involving everyone and sometimes you are thinking "Oh yes we have done the pathway now" and then you realise "oh we missed that one important point. Can we walk through it again?". (HP5)

The magnitude of these changes meant that team-working and communication became even more integral, and many participants described the value of ongoing team discussions with a specific focus on COVID-19-related issues. Technology such as video conferencing software or messaging groups had played a huge role to facilitate this.

We were all social distancing at this point and we were having discussions mainly over Microsoft Teams… it's much better now. We started running our cancer MDT [multidisciplinary team] on Teams too, that was pretty effective, and we managed to get pretty much the same amount of collaboration or potentially even more because it's quite convenient to be able to dial in from anywhere… I think we'll probably continue to use it in some way all the way into the future hopefully. (HP3)

---

> **Box 1  Learning lessons the hard way**
>
> ► Reflecting on optimal personal protective equipment use in patients:
> HP40: We're doing things which may be putting ourselves at more risk than non-risk, because they seem to be sensible at face value, but maybe, are not… We've been asking patients to wear face masks at the time of surgery… as soon as that happened, the lens we were using for the surgery was steaming up every few minutes. So I thought, "Well, if it's steaming up, it must mean that the patient's breath is coming up the mask and out of the drape that we've put on and into the surgical field." So actually, paradoxically, putting a mask on the patient was probably increasing my exposure in the surgical field, rather than decreasing it. But intuitively, everyone thought that if we put a mask on a patient during surgery, we were reducing our risks… So from a protection perspective, we're still learning, I think. There's no doubt that we will be better prepared for the next wave that is being predicted.
> ► Organisational lessons:
> HP4: We turned the top floor into an intensive care unit. And when that started filling up, we realised it was just a really bad idea. Because, although it's a nice idea having patients isolated in separate rooms, if a patient is in a separate room, they need one nurse to look after them in each room. And so, it's very nurse heavy. Whereas, actually, once you start getting a lot of patients, what you need is to be able to cohort the patients. So, 3 or 4 in a bay. And you can have 1 or 2 non-skilled nurses—you know, not ITU-skilled nurses—in there being supervised by 1 ITU nurse. So, you can dilute your skills… The other thing with that plan, the idea was, as our number of patients increased, we could keep them all in the same block and we would just expand intensive care down from the top floor, to the floor below, to the floor below that but unfortunately … there was only an adequate back-up power supply to the top floor. And the next 2 floors below had nowhere near enough back-up power. So, if the mains power went down, we'd kill everyone on the ward instantly. So, we've had to abandon that completely and decant an entire intensive care unit to somewhere else and open a new one up. And build another intensive care unit. Because one of things no-one had thought to check was, actually, was there enough power there.
> ► Optimising pathways:
> HP4: Our plan was always going to be we would collect the patients take them to intensive care, anaesthetise them there and do everything there. But we very swiftly realised that that actually meant a trip with a potentially very unstable patient, through the hospital, not in a controlled situation. We risk contaminating what may be clean bits of the hospital. And so then we actually moved to completely just doing everything on site. So, we essentially take intensive care to the patients. And that saves a lot of time. We think made the process much safer for the patients. And made the process safe for everyone else in the hospital as well. Because people who were infected were only then being moved through the hospital a minimal amount of time, to decrease contamination risk.

Informants reflected on the challenges of having to adapt as best they could with only limited evidence to guide them (box 1). One anaesthetist pointed out the potential implications of ad hoc innovations in this unusual time:

One of the most dangerous things in medicine is what's called the MacGyver bias. Which is where people believe that their ability to, to fudge a solution out of something will be much better than the usual way of doing it. I think you always have to be aware that,

**Table 1** Modifications to practice for intubating patients

| Example | Quotation |
|---------|-----------|
| Changes to mode of anaesthesia | HP40: One of the innovations that has happened as a direct result of COVID is that surgical procedures that, up until now, we would have said were general anaesthetic procedures have been successfully converted to local procedures… We would never have contemplated doing an enucleation—removing the eye—under a local anaesthetic before. Now, we have successfully done that. |
| Changes to where the patient is intubated | HP28: The anaesthetic is being conducted in the theatre whereas normally it would be done in the anaesthetic room in the UK … It just means that they are not moving between lots of different places as well, so they try to minimise the number of points that the patients will stop at. |
| Changes to where patient extubated | HP4: Now we would actually wake the patient up in theatre and have them awake in theatre before they went out, where previously, patients would routinely be taken to the recovery area mostly asleep… In the past, where you might have a patient waking up with an airway in, coughing it out, coughing and spluttering a bit. But actually if they do all that whilst we're in the theatre environment, which is already contaminated, and we're all in protective kit. |
| Reducing aerosolisation during intubation | HP34: If you put a plastic sheet over the head of the patient that reduces much of the aspiration, so the spit, coming out and then the aerosolisation which causes spit will sit on that plastic more than hit you in the face or the neck or anything like that. Those are good things that came out and you can actually see it, the one I saw, you saw all of the spit on the patient, when you extubate, the spit went all over the patient, if you didn't have that plastic sheet, that would be, because your neck is still exposed and all of that. |
| Pausing to allow aerosols to be dissipated | HP17: There's a delay between the patient being put to sleep and the operation being started, just to allow the air to be changed in theatres. The rationale behind that being that when someone's put to sleep and they've got a breathing tube put down, the virus might be kind of put into the air, they might be floating around. Operating rooms have got incredibly high air flow, the air is changed—it depends on your particular operating theatre but each individual hospital will know how long it takes to cycle the air out of their theatre, and then they can do a bit of maths to see how long it takes for the air in there to be entirely fresh. It's normally about 20, 25 minutes. |

whilst it's important to adapt and to—in this sort of situation particularly you're having to innovate and change the way you work and change the way you do and change the way you approach things. Always be aware that, if you're the first person to come up with a way of doing it, the chances are that a lot of other people will have thought of that before and decided it's a bloody stupid idea overall. (HP4)

### Innovations within high-risk specialties

There was agreement that healthcare professionals who participated in procedures within the head and neck region were at increased risk of infection. For instance, anaesthetists were at high risk due to aerosolisation when intubating and extubating patients due to direct access or direct instrumentation of the airway. There were many strategies adopted in these contexts (table 1).

Media coverage showing deaths of ear, nose and throat (ENT) surgeons appeared to have impacted on the surgical community's perception of risk for this group, and many described how it made them scrutinise their own practice. Some of the ways that tracheostomies, for example, were adapted to try to mitigate this risk are outlined in table 2.

Basically the highest concentration of virus in an ill Covid patient is in anything that comes in and out of their mouth and nose, if they've got Covid contains the highest concentration of viral particles with the greatest risk of cross infection to somebody looking

after that patient. Very sadly, the first Doctor to die in the UK was an ENT surgeon and they spend their entire lives examining peoples' upper aerodigestive tract and therefore they were at particularly high risk. (HP10)

### AGPs: complex and uncertain risks

Participants described how new national and local guidelines had stated that AGPs such as laparoscopic and endoscopic approaches subjected staff to elevated risk. However, many reflected that there had been some uncertainty as to precisely what constituted an AGP within their surgical teams. There were also conflicting views as to whether, in line with guidance, alternative techniques should be adopted.

It's really uncertain. We're bombarded by information from national societies, international societies, there's webinars left right and centre, and I've certainly taken part in a fair few of those to try and understand peoples' perspectives. And we've got that sort of headline idea that surgeons are at risk from aerosolised procedures, but we don't really have any knowledge of what an aerosolising procedure is. (HP13)

When contemplating whether to proceed with AGPs, teams weighed up many factors, including the potential risks to the patient, existing evidence (or lack thereof), and what colleagues or those in the wider surgical

**Table 2** Modifications to forming a tracheostomy

| Example | Quotation |
|---|---|
| Being performed only by a small team of experts | HP43: Early on, I was interested in working [on tracheotomies] and a surgeon was interested in it as well. Our personalities work reasonably well together and we started, from the surgical side at least, combining forces and trying to figure out a way to do this safely… We weren't exactly sure which patients were going to make the most sense. We didn't know who exactly would benefit from it, but there was little question there were going to be some people … We go into the room, we wheel in all our stuff. We don't even talk that much anymore. Teamwork is very crucial… I think it has been important, both from an operational standpoint, but also from a mental support and enthusiasm standpoint. I think, if one of us was doing this alone, without the other, I think it would be a lot more difficult. I wouldn't say it's undoable on your own completely, but it's just crazy to try to do that. You have to ask for help. |
| Pausing the ventilator | HP41: I think a tracheotomy is more risk… Before we cut open, cut open the trachea, we make the ventilator pause for a second. Then first we pause the ventilator, then we cut, and then we insert the intubation quickly, and we restart the ventilator … If the ventilator is open, then there are a lot of aerosols… Secretion maybe comes out from the lungs, from the bronchia. So if we pause the ventilator, maybe they can decrease the secretions. |
| Location of tracheotomy | HP41: All the tracheotomy is at the bedside. So it's including a resident, to help at the head side of the patient, and two surgeons. One is at the bedside, to do the surgery. And one or two nurses help to deliver some things to help the surgery. Now, the unit is the ICU unit. There's one person per room. |

ICU, intensive care unit.

community were doing. Taken together, informants' accounts suggested that the issue of AGPs created conflicts between their clinical roles where they had to weigh up what was best for their patients while contemplating their own safety.

It's not all about saving the physician and the team, it's about everybody in the process and understanding that these are really really, difficult decisions… They're not black and white, they're grey… It's about protecting surgeons, it's not about protecting patients… I'm not sure there's been another example where it's been so clear that effectively national guidance has come out to say you put yourself ahead of the patient care, which is how I interpret the guidance that came out. (HP19)

As the quotes below highlight, informants were conflicted as to how comfortable they felt about putting their own safety ahead of the patient's.

We have to accept risks to us because the patient needs an operation far more than we need to be protected… You look at the newspapers and the clinicians and the nursing staff and the porter staff that have all fallen foul of this, people are going to continue to work, there's an inherent dedication to achieving a safe outcome for a patient if that necessarily places you at risk as well. (HP28)

Although you want to do everything for your patient, your own safety and the safety of your team has to come first because if you have a whole team who is not properly protected and everybody gets infected you have taken out a team or an individual of that team for a period of time, so they can't help with the future patients. (HP9)

When proceeding with AGPs, participants described adopting the highest level of PPE and implementing strategies and innovations to try to minimise the spread of the virus during aerosolisation (table 3).

### Adapting to PPE-related challenges

At the time of interviews, most participants felt that the PPE provided was sufficient, although some interviewees expressed their concerns about not having access to adequate PPE and described having to outsource their own or 'pushing' their hospitals to provide more or fuller PPE. Moreover, several informants reflected on the news coverage that there had been limited PPE available to hospital staff, and expressed concerns that this would happen to them as the pandemic progressed. As guidance emerged stating that healthcare professionals should not put themselves at risk if they did not have sufficient PPE, many were again forced to consider whether to put their own safety before their patients':

Bottom line is, I have a wife and children […] I am not going to put my family at risk by doing something without the right kit. Sorry. That's the way it is. If I haven't got the equipment to do it, I'm not doing […] There's national guidelines on what we should wear for each situation. And if we haven't got it, I'm not doing it. (HP4)

Some participants also expressed concern as to whether their PPE was fitted correctly, and teams developed local strategies to facilitate optimal PPE donning/doffing. This included creating visual aids (checklists, signs/posters in theatre), having 'buddy systems', and engaging in team simulations to practise and contemplate different donning/doffing scenarios. Most informants reported that wearing PPE was hot, distracting and uncomfortable,

 Elliott D, et al. BMJ Open 2021;11:e046662. doi:10.1136/bmjopen-2020-046662

**Table 3** Modifications to AGPs

| Examples | Quotation |
| --- | --- |
| Wearing full PPE | HP28: We would use keyhole surgery and we use various adaptations for the keyhole …we did everything with PPE at the time we saw the patient, full PPE in theatre (…) We went against guidance and used keyhole surgery because we felt that there was no evidence to suggest that it's any more risky. |
| Use of smoke-filter evaporators | HP21: We use filters to avoid aerosols, every port is connected to a filter so no aerosol comes through the port… We fix the port to the skin to avoid accidental removal, apply a stitch to the port and fix it to the skin, so the chances of the port being accidentally removed is really decreased. |
| Plastic sheets | HP12: We cover the head and the drill with this plastic sheet that aerosols are not going into the air, and not going everywhere, to protect the environment and to protect the surgeons. |
| Drapes | HP18: We're using some simple things like some plastic drapes over the top of the really aerosol generating stuff that we're doing, making some holes in those drapes so that we can still use our normal instruments. |
| Adopting a combination of approaches | HP20: Tracheostomies were done percutaneously … They have done some modification, doing some combination between open and percutaneous tracheostomy…to minimise as much as possible the risk of leaking aerosol in the operative field to the operative team. |

AGPs, aerosol-generating procedures; PPE, personal protective equipment.

although there was a clear sense that the discomfort was a necessary compromise for sufficient protection.

> I am just constantly thinking, have I done this right? Am I putting this on in the right way? Is my mask fitting properly? I just don't know, and then the real feeling of… I don't want to, I really don't want this on my face during this procedure, and am I doing the right thing. And I think one of the things for me is that I don't really feel like we have had very good instructions on how to do these things. So, one of the things that hit me this morning was, nobody really seems to know, nobody seems to be in control, or what's going to happen and these are the processes, this is where you put your stuff. So when you are putting it on and taking it off etc etc, it is a bit chaotic and that makes me really nervous… I hope I got it right, but I don't know. (HP3)

Wearing PPE in the operating room also created practical problems that had the potential to impact performance during surgery. The time taken to don PPE was considered problematic in emergency circumstances. One strategy for enhancing efficiency was to prepack PPE in 'grab and go' bags to enable staff to proceed more swiftly to theatre. PPE made it significantly harder to communicate between staff in theatre and between staff positioned in 'clean' and 'dirty' areas. Staff were also unable to obtain additional surgical equipment while in the operating theatre. Consequently, many participants described strategies to facilitate communication (table 4).

For some, PPE affected how operations were performed. Surgeons described, for example, how wearing two sets of gloves affected their dexterity and how using a microscope was difficult while wearing a visor. These participants described removing their PPE to complete specific tasks. The issue of surgeons having to weigh up being fully protected by PPE and their ability to operate once again demonstrates the tension between the safety of patients and healthcare professionals.

> The problem surgically with wearing the visors are that the visors sticks out from your face, so I desterilise the instrument on the visor and I desterilise my gloves and the visor so then at the end of the procedure I ended up just having to take the visor off because it was a sterility risk for the procedure I was trying to perform. Trying to use the operating microscope with the visor or with googles on outside of my glasses is just not possible. So at that point, I have areas of my face exposed while I am looking down the operating microscope puttering away at a tumour over creating aerosols whilst not fully covered and I can't be fully covered, else I can't do the surgery if I am. (HP16)

## DISCUSSION
### Summary of findings and lessons learnt
This study captures how surgical teams were suddenly faced with new and unfamiliar risks due to COVID-19. We report how they made sense of and adapted to these risks as the pandemic unfolded. Informants described how there was a plethora of evolving—and sometimes contradictory—local and national recommendations about providing surgical care safely during the pandemic. Our study highlights that guidelines were not a panacea for surgical practice, and at times, there were tensions between clinical roles and personal safety.

We found that communication and teamwork were crucial to how teams adapted. Within this, we identified several practices that appeared helpful for reducing risks and optimising surgical practice during the duration of this pandemic (and potentially other adverse situations). First, healthcare professionals articulated

**Table 4** Problems caused by PPE and potential strategies to overcome these issues

| Issue caused by PPE | Quotation | Potential solution | Quotation |
|---|---|---|---|
| Time taken to don PPE | HP19: It's made the operations much longer, by the time you get the kit on. | 'Grab bags' | HP4: We've set up a fantastic production line of grab bags. So, everything we need is in one of a series of grab bags.(...)We're going off to intubate someone, so we want that bag and that bag. We're going to put lines in them, so we want that bag. |
| Difficulty determining who is who | HP3: You can't take name badges in. | Name and roles written on PPE | HP11: People do write their names on their gowns or aprons.... The problem is you don't know what their role is. They should actually write, 'ITU Nurse'. |
| Difficulty communicating while wearing full PPE | HP11: I could barely hear anybody. It was really hard. | Using equipment to facilitate communication | HP34: Vocera is a communications system that you can have clipped on you or if you're in the scrub role you can have it on the side and you can activate it by clicking it. With the N95 mask you don't hear people. If you have Vocera then you can speak and people can hear. |
| PPE fogging up glasses | HP19: When I was wearing my spectacles when I was breathing out the air was misting up my spectacles. | Masks over glasses | HP20: We have been working on the last week on a modification. You have seen some reports in different countries about the use of a snorkel mask... and it seems to work much better, without fogging. |
| Needing additional equipment | HP3: Normally if we need something extra during the operation, someone just walks out of theatre, get its and then comes back in again, but we are not allowed to do because there are clean areas and dirty areas. | Equipment to aid communication with 'runners' | HP8: Well I brought some walky-talkies which are quite good which help with communicating from inside to outside of the room. So that is one problem. They are quite useful.... the runner outside will have one and we have one in a plastic bag in the theatre. I actually bought them on eBay and everyone wants more now so I am buying some more which should arrive any day.... Shouting through the door or holding up signs is difficult. These work quite well actually.<br>HP3: One of the anaesthetists had the idea of having some Bluetooth speakers, so one in the clean area and one in the dirty area. |
| Long periods wearing PPE | HP34: We realised people can't do nine hours in a theatre with PPE on, wearing the N95 puts incredible pressure on people's faces. | Changes in staffing | HP35: We have to do a team change, so we changed how we did things there as well. We started looking at team changes and how we would manage those, we'd send people home, knowing that in 4 hours, they'd be coming in to do a team change. |

ITU, intensive therapy unit; PPE, personal protective equipment.

the importance of ensuring all aspects of the preoperative, operative and postoperative pathway was carefully mapped out to identify potential risks and staff roles. This enabled teams to minimise contact points and the number of people in the operating theatre to avoid potential infection transmission and contamination. It is important to note that pathway is likely to need to be reviewed frequently, in light of emerging evidence, experiences and concerns. To ensure practice is dynamic and cohesive, several participants in this study highlighted the value in ensuring all members of the surgical team were communicating regularly to discuss and reflect on emerging evidence, guidance, experiences and concerns. Participants in our study also conveyed the importance of working together and developing local strategies to ensure PPE is donned and doffed safely and efficiently. This included visual aids (checklists, signs/posters in theatre), having 'buddy systems' and engaging in team simulations.

## Implications of findings

Surgical teams sought to mitigate the risks by modifying their current practice with multiple strategies and innovations to deliver patient care that they felt comfortable with. Nonetheless, uncertainties remained about optimal strategies to minimise risks. Although the epidemiology of COVID-19 has been well reported, there is little robust evidence regarding safe and best surgical practice.[38] Our study supports an urgent need for high-quality multicentre research,[17] particularly as the pandemic continues to disrupt health services around the world. It has been argued that with the constraints of time and resource, clinical practice must be driven by a pragmatic and novel approach to collate global evidence and to inform evidence-based practice.[38]

This study has also highlighted that COVID-19 has had a significant impact on healthcare professionals. This includes experiences of fear and anxiety about personal safety, the challenges of making sense of multiple guidelines that changed frequently, an evolving workload and wearing more PPE. This is supported by findings from interviews with 14 surgeons from a hospital in Ireland, which reported how COVID-19 had fundamentally impacted clinical roles and had consequences on wellbeing.[39] It has been argued that the surgical community needs to be provided with support to limit the inevitable psychological burden and risk of burnout.[40–44] Shanafelt and colleagues also emphasise that visible leadership is crucial during this turbulent time, and that leaders will need to establish innovative ways to engage with their teams to fully listen to and understand the sources of concern, and work with them to develop approaches that mitigate concerns.[45] Interventions must determine how best to support healthcare professionals to process and navigate the physical and psychological repercussions of the pandemic.

## Strengths and limitations

The study used qualitative methodology to provide new insights into perceived risks in surgery in the context of COVID-19. Measures were undertaken to ensure rigour of the data in ensuring findings reflected the meaningful impact of COVID-19 on surgical teams. Rigour in qualitative research is determined by a range of criteria.[31] Having multiple experienced qualitative methodologists conducting the interviews, analysis and write-up enabled analyst triangulation to strengthen credibility of the study.[31 37 46] To ensure consistency in analysis, one individual acted as an 'analytical auditor' and reviewed each transcript to enable strategic overview of the study findings and review the data for discrepancies or overstatements.[30 33 46] Emerging themes, with reference to the raw data, were discussed with wider members of the study team (including academic surgeons), as well as research participants who were not part of the study team, to ensure a good fit between researchers' interpretations and representation of the participants' experiences.[30 31 47]

Our study had some methodological limitations. Initially, individuals who were known to the study team were approached to consider participation and subsequent participants were identified via snowball sampling. It is therefore possible that the findings may not be representative of all healthcare professionals. The researchers in this study were known to have a particular interest in surgical innovation and responses may have also been influenced by participants' perceptions of what might constitute socially desirable answers, particularly given the emotive nature of some of the topics discussed in the interviews (eg, resource constraint, balancing risks to patients vs risks to themselves). However, contrasting perspectives did emerge from our sample, and we sought to search for and report negative cases that did not align with recurring ideas. The findings have been reported with negative cases in mind, to present as comprehensive an account of key themes as possible.[22]

It is important to note that interviews were conducted between March and May 2020, which has limitations for the transferability of results. While we were able to capture initial responses as the pandemic unfolded in its first wave, future research is warranted to explore how teams continue to adapt during the later stages and aftermath of the pandemic. There are also limitations in the diversity of countries and professions represented in this study, as the majority of participants were from the UK (76%) and surgeons (83%). Findings may have been further developed with greater representation of non-UK settings, and other professions (eg, anaesthetists, nurses, operating department practitioners). Moreover, participants were from developed countries. Risk mitigation in surgery may look very different in low-income and middle-income countries with fewer resources, different healthcare infrastructures and more recent experience of epidemics. Capturing a wider range of international perspectives would inform the wider discourse on optimal practices for surgical procedures during the pandemic.

## CONCLUSION

In summary, our study provides insights into how surgical teams had to rapidly modify their practice with a wide range of innovations to mitigate the risk of COVID-19. Communication and teamwork played an integral role in how teams adapted. While an abundance of guidance was available, perceptions of risks were complex and context dependent. Taken together, this study highlights that the context of surgery has changed dramatically and that there are many uncertainties about best practice. As the COVID-19 pandemic continues to unfold, it is imperative that future research continues to explore how teams adapt and that surgical practice is based on high-quality evidence that optimises the safety of both patients and healthcare professionals.

**Author affiliations**
[1]National Institute for Health Research Bristol Biomedical Research Centre, Surgical Innovation Theme, Centre for Surgical Research, University of Bristol, Bristol, UK
[2]Population Health Sciences, Bristol Medical School, University of Bristol, Bristol, UK
[3]University Hospitals Bristol and Weston NHS Foundation Trust, Bristol, UK
[4]Division of Neurosurgery, Department of Clinical Neuroscience, University of Cambridge, Cambridge, UK
[5]Leeds Institute of Medical Research, Level 7 Clinical Sciences Building, St. James's University Hospital, Leeds, UK
[6]Ear Institute, Faculty of Brain Sciences, University College London, London, UK

**Acknowledgements** We would like to thank Rosetrees Trust for funding this study, and each participant for taking the time to share their experiences with us. We are also grateful to Murat Akkulak, Steven Beech, Becky Carthy, Jenny Russe, Jane Collingwood, Chris Pawsey and Tom Steart-Feilding for providing help and support for the study.

**Contributors** DE was the project lead and developed the study protocol, with all authors reviewing and providing comments on subsequent versions. DE, CO, LR, MJ, KA, SP and JD conducted interviews. DE, CO, LR, MJ and JD analysed the data. JB, MB, NSB, PJH, DJ, DE, MJ, LR and JD identified potential study participants. DE drafted the initial version of the manuscript, with critical revisions from JD, JB and LR. JB, MB, NSB, DJ and PJH conceptualised the study design. All authors commented on the manuscript and gave approval for the manuscript to be submitted. The NIHR Bristol Biomedical Research Centre Surgical Innovation theme is directed by JB (theme lead).

**Funding** This study was funded by the Royal College of Surgeons (RCS Covid Research Group) and Rosetrees Trust, with support from the National Institute for Health Research (NIHR) Biomedical Research Centre at University Hospitals Bristol and Weston NHS Foundation Trust and the University of Bristol and NIHR Advanced Surgical Technologies Incubator. JD and JB are NIHR senior investigators. NSB is an MRC clinician scientist.

**Patient consent for publication** Not required.

**Ethics approval** Ethical approval was granted by the University of Bristol Faculty of Health Sciences Research Ethics Committee (reference: 56522).

**Data availability statement** No data are available.

**ORCID iDs**
Daisy Elliott http://orcid.org/0000-0001-8143-9549
Natalie S Blencowe http://orcid.org/0000-0002-6111-2175
Kerry NL Avery http://orcid.org/0000-0001-5477-2418
Peter Hutchinson http://orcid.org/0000-0002-2796-1835
Jane M Blazeby http://orcid.org/0000-0002-3354-3330

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
