## [Reviewer comments · BMJ Open]

ARTICLE DETAILS

TITLE (PROVISIONAL)	“Overnight, things changed. Suddenly, we were in it”: A qualitative study exploring how surgical teams mitigated risks of COVID-19
AUTHORS	Elliott, Daisy; Ochieng, Cynthia; Jepson, Marcus; Blencowe, Natalie; Avery, Kerry; Paramasivan, Sangeetha; Cousins, Sian; Skilton, Anni; Hutchinson, Peter; Jayne, David; Birchall, Martin; Blazeby, Jane; Donovan, Jenny; Rooshenas, Leila

VERSION 1 – REVIEW

REVIEWER	Dale F. Whelehan Discipline of Surgery, School of Medicine, Trinity College Dublin, Ireland
REVIEW RETURNED	20-Nov-2020

GENERAL COMMENTS	Thank you to the author and their team for the opportunity to review this important and timely research. I commend the researchers for their rapid turnaround of the research to deliver some prompt findings. I thoroughly enjoyed reading the manuscript. Regarding my review, I have comments for the researchers to consider and to respond to. In particular, I recommend the author focus on elaborating on the methodology to show sufficient rigour in the design and implementation of the studies data collection and data analysis processes. For the purpose of the review I have highlighted some areas for the author to consider below. Abstract: [ ] Differentiate between results, discussion and conclusions[ ] There is no conclusion section within the text Page 5: [ ] Rephrase the sentence starting with “innovation can be associated with..” into two sentences[ ] Were ethical considerations not needed for research conducted outside the United Kingdom – please explain this and if there was data sharing agreements for GDPR reasons[ ] Remove the PPI section – not relevant Page 6: [ ] The sampling strategy: how did you protect against selection bias? Please differentiate how you recruited different cohorts more explicitly in your sampling strategy[ ] Please provide evidence of qualitative experience of the researchers
---

- 'grounded theory methodology' – please discuss further your interpretation – was it constructivist grounded theory? Mention of thematic analysis elsewhere
- What type of coding framework were you using – were you focused on content, feeling/affect, perceptions etc. ?
- How did you determine whether to include negative cases in your findings or not – not as clear
- I would like to see reference to qualitative metrics of rigour used in the study i.e. trustworthiness of the data

Page 7:

- How do you control for differences in interviewing style given there were several completing interviews? Was there a standard setting for this approach
- Discuss how you you were reflexive in your research and minimised personal perceptions in theme formation
- '9 individuals didn't reply' – what was your sampling frame? How do you determine this if you used a few sampling strategies?
- 'individuals in nursing roles' – can you clarify if you mean nurses only here?
- A breakdown of demographic numbers would help with insight of the findings i.e. how many within specialties etc.
- it would be good here to indicate what your desired 'representative sample' was. You spoke about purposive sampling for representativeness. Can you elaborate on demographic breakdown of specialties, gender breakdown within specialties, and was 77% male reflective of the 'surgical team' ?

Page 8:

- 'interviewees reflected on the shock ...new risks to their own health' - this is an interesting insight – was there more 'bravado' within some professions more than others regarding contracting the virus compared to other professions. Can you explain more into differences between surgical team members?
- 'the changing context of surgery' - i think this theme is valid - but given it is under 'risk' it might be better to reframe in the context of there being a 'risk associated with practice changes due to COVID-19' - I think this is what the author is explaining but making it more explicit would be useful.

Page 9:

- Table 1: it would be nice to 'categorise' these learning lessons i.e. learnings within the operating theatre ; learnings with new machinery ; learnings in postoperative care etc. Also what were the actual 'lessons learned'

Page 10:

- Remove repetition in text and table headings – on page 10 and relevant for all tables
- Would quote HP40 fit better within Table 2? –

Page 11:

- Navigating uncertainty and complex decision: a relevant theme again - but since it is in light of the heading 'risk' might it be useful to frame it in the context of how individuals found themselves within 'risky' situations due to the 'uncertain and complex decisions' in which they had to navigate in? I think it relates it back to the main heading much clearer this way.

Discussion:

	[ ] 'leaders will need to establish...' - what suggestions do you have regarding supporting HCW in this regard? Did you participants offer you any advice on how best to tackle the ever growing changes withing organisations? Did you sample much 'senior leadership' within organisations and did they suggest anything useful? This might be worth considering as a follow-up study [ ] I appreciate the recommendation around required research in developing countries responses to COVID-19 - a very valid and important point to make. [ ] 'representative of all healthcare' - or even surgical teams - I would have liked to see some discussion around differences in approaches between the countries and professions. For example, were some cohorts more/less affected by the emotional aspects? How did countries differ in their approaches? There were some significant differences in impacts between Italy and the United Kingdom and recognition of this would be important within the discussion. [ ] you include a reference from Guba (28) but don't explain what it refers to explicitly - please explicitly mention in text if using reference. [ ] Have you any particular recommendations around communication and teamwork management as these variables played a significant role in to how teams adapted? Reporting Checklist: [ ] Make more explicit the objectives of the research (i.e. aims and objectives) [ ] 'qualitative approach' isn't clear from the methodology - would appear that it fell into the paradigm of constructivism. Was constructivist grounded theory [ ] 'researcher characteristics' - I do think explaining more about who the researchers are and how they are situated within the environment they are researching is important. Do they know the participants? Were interviews done in person? in what setting? [ ] 'setting; - were the settings public? one thing i noticed missing was the cancellation of many elective surgeries - did this feature within the findings? did individuals perceive this to be a 'risk' to their level of training/competency? [ ] 'techniques to enhance trustworthiness' - this would be important to reference more. This article might be helpful - Irene Korstjens & Albine Moser (2018) Series: Practical guidance to qualitative research. Part 4: Trustworthiness and publishing, European Journal of General Practice Well done again to the authors on this study. I look forward to seeing your responses. Regards, Dale F. Whelehan
--	--

REVIEWER	Molly Jarman Brigham and Women's Hospital, Boston, Massachusetts, USA
REVIEW RETURNED	05-Jan-2021

GENERAL COMMENTS	Thank you for the opportunity to review this manuscript reporting findings from an international qualitative study of surgical care during the first wave of the COVID-19 pandemic. Introduction focuses on guidelines/practices in the UK, but study is international in scope - the intro would be strengthened with some mention of variation in guidelines across national systems. Given variation in guidelines/policies across health systems, did interviewers modify questions/probes to address differences at sites outside the UK? For example, how PPE shortages in the US, or extraordinary stay at home orders in New Zealand? The findings of this study are valuable as we move through the second wave of the current pandemic, but perhaps more valuable in relation to future pandemic disease. The paper would benefit from some discussion of implications of "lessons learned" for future pandemic or other disasters. Similarly, how might these findings impact surgical care in non-pandemic/disaster settings? Are any of the challenges discussed likely to persist after the pandemic subsides?
---

REVIEWER	Neal Stephen Kleiman Houston Methodist Hospital
REVIEW RETURNED	10-Jan-2021

GENERAL COMMENTS	The authors sampled a variety of individuals involved in the care of patients during the COVID era, predominantly in surgical specialties, and report their responses to an interview and discussion concerning the operational and emotional issues posed by the pandemic. Overall, the responses are fascinating to read. However, in terms of scientific study, there seems to be a tremendous amount of subjectivity here. The interviews may have been structured, but it's unclear how similar one interview was to another, whether the subjects were informed that the responses were going to be stratified into themes, and how the reported responses were selected. To this reviewer, the style is very reminiscent of a Ken Burns documentary (very popular on this side of the Atlantic) -- absolutely fascinating, extremely informative, often poignant -- but obviously subject to selection bias in terms of choosing exactly what to report. These caveats must be elucidated in the Discussion. The Methods section should accordingly describe to the best of the authors' ability, the criteria used to select the quotations that are included.
---

VERSION 1 – AUTHOR RESPONSE

Reviewer 1: Dr. Dale Whelehan, Trinity College Dublin School of Medicine

Comments to the Author: Thank you to the author and their team for the opportunity to review this important and timely research. I commend the researchers for their rapid turnaround of the research to deliver some prompt findings. I thoroughly enjoyed reading the manuscript. Regarding my review, I have comments for the researchers to consider and to respond to. In particular, I recommend the

author focus on elaborating on the methodology to show sufficient rigour in the design and implementation of the studies data collection and data analysis processes. For the purpose of the review I have highlighted some areas for the author to consider below.

Response: Thank you for your positive and constructive feedback. Please see the details of the changes made below.

Abstract:

-Differentiate between results, discussion and conclusions

Response: 'Results' and 'Conclusions' sections are separated and signposted within the abstract. We have structured the abstract to comply with the journal guidelines (which state that there should not be a specific 'Discussion' subheading), although we believe that the 'Conclusions' section captures points raised in the discussion section too:

Results: *Surgical teams sought to mitigate COVID-19 risks by modifying their current practice with an abundance of strategies and innovations. Communication and teamwork played an integral role in how teams adapted, although participants reflected on the challenges of having to improvise in real time. Uncertainties remained about optimal surgical practice and there were significant tensions where teams were forced to balance what was best for patients whilst contemplating their own safety.*

Conclusions: *The perceptions of risks during a pandemic such as COVID-19 can be complex and context dependent. Management of these risks in surgery must be driven by evidence-based practice resulting from a pragmatic and novel approach to collation of global evidence. The context of surgery has changed dramatically, and surgical teams have developed a plethora of innovations. There is an urgent need for high-quality evidence to inform surgical practice that optimises the safety of both patients and healthcare professionals as the COVID-19 pandemic unfolds.*

- There is no conclusion section within the text

Response: Thank you for raising this. A 'Conclusion' section has now been demarcated on page 19:

"Conclusion

In summary, our study provides insights into how surgical teams had to rapidly modify their practice with a wide range of innovations to mitigate the risk of COVID-19.

Communication and teamwork played an integral role in how teams adapted. Whilst an abundance of guidance was available, perceptions of risks were complex and context dependent. Taken together, this study highlights that the context of surgery has changed dramatically and that there are many uncertainties about best practice. As the COVID-19 pandemic continues to unfold, it is imperative that future research continues to explore how teams adapt and that surgical practice is based upon high-quality evidence that optimises the safety of both patients and healthcare

professionals.”

Page 5: Rephrase the sentence starting with “innovation can be associated with..” into two sentences

Response: We have re-phrased this sentence to enhance clarity:

“Innovation could involve: modifications to decisions around whether and how to operate, surgical techniques, tools and technology, surgical team composition and operating conditions (11).”

- Were ethical considerations not needed for research conducted outside the United Kingdom – please explain this and if there was data sharing agreements for GDPR reasons

Response: Thank you for checking. An ethics amendment was processed to enable us to conduct data collection outside of the UK. This has now been clarified on page 6:

“Ethical approval was granted by the University of Bristol Faculty of Health Sciences Research Ethics Committee (reference: 56522), and an amendment enabled data collection to be conducted via University-approved video conferencing software and for interviews to be conducted internationally.”

Because there were no plans to share the raw data, and because participants were giving their own experiences of the pandemic and not speaking on behalf of their institutions, it was determined that no data sharing agreements were required.

- Remove the PPI section – not relevant

Response: Thank you, section deleted as advised.

Page 6:

- The sampling strategy: how did you protect against selection bias? Please differentiate how you recruited different cohorts more explicitly in your sampling strategy

Response: We have added further information about the sampling strategy (updated text in bold) on page 6:

*“Healthcare professionals involved in the management and treatment of patients undergoing surgical procedures were eligible. A key informant sampling approach was initially adopted(18), whereby individuals **from different surgical specialities** who were known to the study team were approached. Subsequent participants were identified via snowball sampling(18), whereby interviewees recommended potential participants. **Participants were also identified through convenience sampling (19)**, whereby study details were circulated to members of surgical groups (the National Institute for Health Research Surgical MedTech Co-operative and the Royal College of Surgeons of England). **The adverts included instruction to contact the***

study team if individuals were interested in study participation. As data collection continued, sampling became increasingly purposeful with a view to achieving a sample of maximum variation, to ensure insights were captured from a range of informants operating in different contexts (19). We aimed to capture variation in relation to role (including surgeons, theatre nurses and anaesthetists), specialty, years in profession, gender and geographic location. Capturing an international perspective was important to identify how best practice evolved, particularly in those areas/nations that were 'ahead of the curve'. **A database of participants and their characteristics was maintained and assessed as the study progressed,** and interviews with those who were underrepresented were prioritised.

As discussed in our response to the Associate Editors comments, we have also reflected on the implications of adopting these sampling strategies in the revised discussion section (page 19).

“Strengths and limitations

Our study had some methodological limitations. Initially, individuals who were known to the study team were approached to consider participation and subsequent participants were identified via snowball sampling. It is therefore possible that the findings may not be representative of all healthcare professionals. The researchers in this study were known to have a particular interest in surgical innovation and responses may have also been influenced by participants' perceptions of what might constitute socially desirable answers, particularly given the emotive nature of some of the topics discussed in the interviews (e.g. resource constraint, balancing risks to patients versus risks to themselves). However, we were able to capture a range of perspectives from surgical teams to explore negative cases that contradicted with emerging findings to understand participants' divergent views and to enhance the transferability of the results (22).

- Please provide evidence of qualitative experience of the researchers 'grounded theory methodology'

Response: A Supplementary File, describing each of the researcher's backgrounds and qualitative research experience, has been created (see Supplementary File 2). This includes information on the qualitative research team, each individual's background and role, and examples of analysts' publications using grounded theory methodology:

“Researcher profiles

All individuals who conducted the interviews (DE, CO, LR, MJ, JLD, SP, KNLA) have extensive experience of the application of qualitative research methods to improve the design and conduct of health services research, and each have PhDs in health-related fields using qualitative methods. All researchers are based at the University of Bristol, and are members of the QuinteT research group (which uses qualitative research methods to optimise recruitment and informed consent to randomised

controlled trials, many of which are surgical) and of the Centre for Surgical Research (which aims to improve the surgical evidence base, and subsequently patient care, through high quality multidisciplinary research). JLD is a Professor of Social Medicine. DE is a Research Fellow in Qualitative Methodology Research. LR is a Senior Lecturer in Qualitative Health Sciences. MJ is a Senior Lecturer in Qualitative Health Science and Senior Research Fellow. CO is a Senior Research Associate in Health Services Research. KNLA is a Senior Lecturer in Health Services Research. SP is a Research Fellow in Qualitative Methodology Research. All researchers involved in the analysis (DE, CO, LR, MJ, JLD) have experience of conducting grounded theory methodology in multiple projects and have published their findings in peer reviewed journals.”

– please discuss further your interpretation – was it constructivist grounded theory? Mention of thematic analysis elsewhere

Response: Thank you for this considered point. We have given this careful thought and do not feel we can label our study as a ‘constructivist’ or ‘classic’ Grounded Theory study, as it did not fulfil the tenets of either approach. On balance, our approach is probably closer to ‘classic Grounded Theory’, but deviates from this approach in several regards. In their original description of GT, Glaser and Strauss did not define a particular epistemological or ontological stance; rather, they presented a series of systematic tools and techniques to conduct social enquiry. We did not ascribe to a full GT approach, as we were not necessarily seeking to develop new theory and therefore we not concerned with development of a ‘core category’. We took a pragmatic approach in that we adopted what we perceived to be the most appropriate methodological tools to address our field of enquiry/aim, which was to capture and represent health care professionals’ interpretations and experiences of surgical practice during a pandemic. We set out to accurately present a comprehensive understanding of these stakeholders’ realities and interpretations, through the devices of interviews and inductive analysis. The intention behind the analysis was to produce key findings that were grounded in the data, as is consistent with classic and subsequent adaptations of GT. To achieve these, we used the constant comparison technique, and employed several approaches to safeguard against the researchers’ personal knowledge and experiences detracting from the participants’ voices (e.g. double coding and discussion, regular analysis meetings, seeking out deviant cases). We have now added the following on to further clarify this:

“Codes were identified inductively using methods derived from grounded theory methodology (20, 21) so that findings were grounded in the data to capture and represent health care professionals’ interpretations and experiences of surgical practice during a pandemic. Through the constant comparison technique, the codes were categorised into themes which were continually compared with the data.” (Page 7)

- What type of coding framework were you using – were you focused on content, feeling/affect, perceptions etc. ?

Response: The data was coded inductively, so that participants' verbal responses in the transcripts were coded based on their contents and the codes were constantly compared to the contents of the other transcripts. We have updated the text in the manuscript to reflect this:

*Transcripts were imported into NVivo (QSR International, USA), where the **contents of the transcripts were iteratively coded** by three coders (MJ, LR and CO). **Codes were identified inductively using methods derived from grounded theory methodology(22, 33), so that findings were grounded in the data to capture and represent health care professionals' interpretations and experiences of surgical practice during a pandemic. Through the constant comparison technique, the codes were categorised into themes which were continually compared with the data. In this way, similarities and differences can be identified and emerging themes refined through the ongoing assimilation of data. (Page 7)***

- How did you determine whether to include negative cases in your findings or not – not as clear

Response: Thanks for raising this. We have now included the following justification for including negative cases in our findings:

“A premise of grounded theory is that these comparisons will enable identification of ‘negative cases’ (i.e. participants whose perspectives/experiences differed from the main body of evidence) (23, 24). This was felt to be important for enhancing the credibility of the analysis as it helped to ensure the findings represented as full and comprehensive an account of participants’ experiences and perspectives as possible, rather than those views that were dominant or fit with a particular impression of the results(20)” (Pages 7/8)

- I would like to see reference to qualitative metrics of rigour used in the study i.e. trustworthiness of the data

Response: We agree this is important point. In line with your above suggestion, we have added the following:

“A premise of grounded theory is that these comparisons will enable identification of ‘negative cases’ (i.e. participants whose perspectives/experiences differed from the main body of evidence) (23, 24). This was felt to be important for enhancing the credibility of the analysis as it helped to ensure the findings represented as full and comprehensive an account of participants’ experiences and perspectives as possible, rather than those views that were dominant or fit with a particular impression of the results(20)” (Pages 7/8)

Reflexivity is an fundamental part of ensuring the transparency and quality of qualitative research(30). Interviewers' reflexive notes, which took into account their observations, thoughts and ideas(31), were shared with the analysis team. Regular team meetings also enabled the interviewers to reflect on the novel process of conducting virtual interviews (a form of functional reflexivity, whereby researchers give critical attention to the way processes influenced research(32)). (Page 7)

We have also now included a section on 'Strengths and limitations' in the discussion section:

This study used qualitative methodology to provide new insights into perceived risks in surgery in the context of COVID-19. Measures were undertaken to ensure rigour of the data in ensuring findings reflected the meaningful impact of COVID-19 on surgical teams. Rigour in qualitative research is determined by a range of criteria (31). Having multiple experienced qualitative methodologists conducting both the interviews, analysis and write-up (30) enabled analyst triangulation to strengthen credibility of the study (31, 37, 46). To ensure consistency in analysis one individual acted as an 'analytical auditor' and reviewed each transcript to enable strategic overview of the study findings and review the data for discrepancies or overstatements (30, 33, 46). Emerging themes, with reference to the raw data, were discussed with wider members of the study team (including academic surgeons), as well as research participants who were not part of the study team, to ensure a good fit between researchers' interpretations and representation of the participants' experiences (30, 31, 47).

Our study had some methodological limitations. Initially, individuals who were known to the study team were approached to consider participation and subsequent participants were identified via snowball sampling. It is therefore possible that the findings may not be representative of all healthcare professionals. The researchers in this study were known to have a particular interest in surgical innovation and responses may have also been influenced by participants' perceptions of what might constitute socially desirable answers, particularly given the emotive nature of some of the topics discussed in the interviews (e.g. resource constraint, balancing risks to patients versus risks to themselves). However, contrasting perspectives did emerge from our sample, and we sought to search for and report negative cases that did not align with recurring ideas. The findings have been reported with negative cases in mind, to present as comprehensive an account of key themes as possible (22).

It is important to note that interviews were conducted between March and May 2020, which has limitations for the transferability of results. Whilst we were able to capture initial responses as the pandemic unfolded in its first wave, future research is

warranted to explore how teams continue to adapt during the later stages and aftermath of the pandemic. There are also limitations in the diversity of countries and professions represented in this study, as the majority of participants were from the UK (76%) and surgeons (83%). Findings may have been further developed with greater representation of non-UK settings, and other professions (e.g. anaesthetists, nurses, operating department practitioners). Moreover, the majority of surgeons who participated in this study were from developed countries. Risk mitigation in surgery may look very different in low- and middle-income countries with fewer resources, different healthcare infrastructures and more recent experience of epidemics. Capturing a wider range of international perspectives would inform the wider discourse on optimal practices for surgical procedures during the pandemic. (Pages 18/19)

Page 7:

- How do you control for differences in interviewing style given there were several completing interviews? Was there a standard setting for this approach

Response: Thank you for raising this question. We have now clarified that the interviews were semi-structured, so a topic guide ensured that the same key areas were covered (while allowing for probing of issues raised by participants). Regular team meetings were held to discuss the topic guide, its suitability and any changes that interviewers felt were required to enhance data capture and to answer the research question. To clarify this, we have updated the term 'discussion' with 'semi-structured interviews' to specify the consistency in interviewing by the different researchers:

Semi-structured interviews were directed by a topic guide to ensure that the same core areas were consistently covered amongst the team of interviewers, whilst allowing flexibility to pursue the detail that was salient to each participant (Supplementary File 3). Specific questions were open-ended to encourage the participant to talk about their own experiences of particular topics ("In your own words, can you talk me through how PPE has been used during COVID-19?"), experiences ("Can you describe a case where COVID-19 affected what happened in the operating theatre?") and views ("In your opinion what are the key risks in surgery due to COVID-19?"). Interviewers were then able to incorporate the interviewee's own terms and concepts into subsequent questions to follow up on specific issues raised and check the interviewer had understood correctly (29). At the end of the interview, participants were given an opportunity to raise relevant issues that had not already been covered (30). Regular team meetings allowed for the

team to review the topic guide in light of findings and consider potential changes (e.g. addition of topics or rephrasing of questions). (Pages 6/7)

- Discuss how you were reflexive in your research and minimised personal perceptions in theme formation

Response: We hope that by including the information on each of the researcher's backgrounds and roles, we have made the researchers more visible in the research paper (1). We have also added in more information about how interviewers reflected on their role in the study:

Reflexivity is an fundamental part of ensuring the transparency and quality of qualitative research(30). Interviewers' reflexive notes, which took into account of their observations, thoughts and ideas(31), were shared with the analysis team. Regular team meetings also enabled the interviewers to reflect on the novel process of conducting virtual interviews (a form of functional reflexivity, whereby researchers give critical attention to the way processes influenced research(32)). (Page 7)

We have highlighted how we conducted various 'quality control' processes to ensure we minimised personal perceptions in theme formation:

The coders met regularly to compare their coding and agree on the main broad categories encapsulating all codes. A subset of the transcripts, batched according to coder, were independently double coded by DE to determine broader consistency in coding approaches (26). Descriptive reports were written on key themes ('Guidance', 'Risk', 'In-theatre processes', 'Organisational', 'Service provision', 'Personal protective equipment (PPE)', 'Testing' and 'Physical and mental impact'). [...] producing descriptive reports provided an opportunity to demonstrate where there was overlap in key findings. Two coders (CO, MJ) also produced a summary on one theme, to explore any major differences in interpreting codes. As multiple qualitative researchers conducted the interviews, one researcher (DE) read all transcripts and synthesised the findings to ensure consistency in analysis and to enable strategic oversight of the findings(26). Results, with reference to the raw data, were discussed with members of the wider study team (including academic surgeons) and several research participants who were not part of the study team. (Page 8)

We have also now added the following information about how we ensured credibility in the analysis:

To ensure consistency in analysis one individual acted as an 'analytical auditor' and reviewed each transcript to enable strategic overview of the study findings and review the data for discrepancies or overstatements (30, 33, 46). Emerging themes, with reference to the raw data, were discussed with wider members of the study team (including academic surgeons), as well as research participants who were not part of

the study team, to ensure a good fit between researchers' interpretations and representation of the participants' experiences (30, 31, 47). (Pages 18/19)

- '9 individuals didn't reply' – what was your sampling frame? How do you determine this if you used a few sampling strategies?

Response: Thanks for raising this. As we stated, recruitment was conducted through emailing known and referred surgical clinicians, as well as having an invitation sent out to surgical mailing groups (National Institute for Health Research Surgical MedTech Co-operative and the Royal College of Surgeons of England). We should have clarified that of those directly emailed by the team, nine were not able to be interviewed. This has now been updated on page 8 as:

“Of the clinicians contacted by the team, nine individuals did not reply to the study invite, said that they were not on the frontline or were unable to find the time for an interview.” (Page 8)

- 'Individuals in nursing roles' – can you clarify if you mean nurses only here?

Response: The term 'nursing roles' was used to encapsulate different job roles among the participants in the nursing department including: practice educators, nurse managers, matrons and charge nurses. We have now clarified this on page 8.

- A breakdown of demographic numbers would help with insight of the findings i.e. how many within specialties etc.

Response: A breakdown of numbers has now been added on page 8:

“Surgical specialties included general surgery (n=14), otolaryngology (n=11), neurosurgery (n=5), cardiothoracic surgery (n=2) and ophthalmology (n=1).”

- It would be good here to indicate what your desired 'representative sample' was. You spoke about purposive sampling for representativeness. Can you elaborate on demographic breakdown of specialties, gender breakdown within specialties, and was 77% male reflective of the 'surgical team' ?

Response: We have now added a point to the discussion:

There are also limitations in the diversity of countries and professions represented in this study, as the majority of participants were from the UK (76%) and surgeons (83%). Findings may have been further developed with greater representation of non-UK settings, and other professions (e.g. anaesthetists, nurses, operating department practitioners). Moreover, the majority of surgeons who participated in this study were from developed countries. Risk mitigation in surgery may look very different in low- and middle-income countries with fewer resources, different healthcare infrastructures and more recent experience of epidemics. Capturing a wider range of international perspectives would inform the wider discourse on optimal practices for surgical procedures during the pandemic. (Page 19)

Page 8:

- 'Interviewees reflected on the shock ...new risks to their own health' - this is an interesting insight – was there more 'bravado' within some professions more than others regarding contracting the virus compared to other professions. Can you explain more into differences between surgical team members?

Response: Among the different surgical team members, there were no apparent specialty-related patterns in informants' perceptions of risk and consequences of contracting the virus. A sentence has been inserted on page 8 to reflect this:

“This was described across a range of specialities and roles.”

- 'The changing context of surgery' - i think this theme is valid - but given it is under 'risk' it might be better to reframe in the context of there being a 'risk associated with practice changes due to COVID-19' - I think this is what the author is explaining but making it more explicit would be useful.

Response: Thank you for the helpful suggestion. We agree. This has now been changed (“Changes to the context of surgery due to COVID-19 risk”).

Page 9:

- Table 1: it would be nice to 'categorise' these learning lessons i.e. learnings within the operating theatre ; learnings with new machinery ; learnings in postoperative care etc. - -- Also what were the actual 'lessons learned'

Response: Thank you for the suggestion. We have inserted a description of the quote to categorise the lessons learnt in Table 1. We have also added the following to the discussion to summarise lessons learned from the study:

We found that communication and teamwork were crucial to how teams adapted. Within this, we identified several practices that appeared helpful for reducing risks and optimising surgical practice during the duration of this pandemic (and potentially other adverse situations). Firstly, health care professionals articulated the importance of ensuring all aspects of the preoperative, operative and postoperative pathway was carefully mapped out to identify potential risks and staff roles. This enabled teams to minimise contact points and the number of people in the operating theatre to avoid potential infection transmission and contamination. It is important to note that pathway is likely to need to be reviewed frequently, in light of emerging evidence, experiences and concerns. To ensure practice is dynamic and cohesive, several participants in this study highlighted the value in ensuring all members of the surgical team were communicating regularly to discuss and reflect upon emerging evidence, guidance, experiences and concerns. Participants in our study also conveyed the importance of working together and developing local strategies to ensure PPE is donned and doffed safely and efficiently. This included visual aids (checklists,

signs/posters in theatre), having 'buddy systems' and engaging in team simulations.
(Page 17)

Page 10:

- Remove repetition in text and table headings – on page 10 and relevant for all tables

Response: Text summarising the quotes/table headings has now been removed.

- Would quote HP40 fit better within Table 2? –

Response: Thank you for this suggestion, this has now been moved to Table 2.

Page 11:

- Navigating uncertainty and complex decision: a relevant theme again - but since it is in light of the heading 'risk' might it be useful to frame it in the context of how individuals found themselves within 'risky' situations due to the 'uncertain and complex decisions' in which they had to navigate in? I think it relates it back to the main heading much clearer this way.

Response: We have now updated this to 'Aerosol Generating Procedures: Complex and uncertain risks'.

Discussion:

- 'Leaders will need to establish...' - what suggestions do you have regarding supporting HCW in this regard? Did you participants offer you any advice on how best to tackle the ever growing changes withing organisations? Did you sample much 'senior leadership' within organisations and did they suggest anything useful? This might be worth considering as a follow-up study

Response: Although not this focus of this research, we agree that it would make a great follow-up study. We have added the following:

Interventions must determine how best to support healthcare professionals to process and navigate the physical and psychological repercussions of the pandemic. (Page 18)

- I appreciate the recommendation around required research in developing countries responses to COVID-19 - a very valid and important point to make.

Response: Thank you.

- 'Representative of all healthcare' - or even surgical teams - I would have liked to see some discussion around differences in approaches between the countries and professions. For example, were some cohorts more/less affected by the emotional aspects? How did countries differ in their approaches? There were some significant differences in impacts between Italy and the United Kingdom and recognition of this would be important within the discussion.

Response: As the purpose of the study was not to compare the responses of each country and the current sample was not diverse enough to do this, we have removed the point about exploring variability between countries in the discussion. We have also added the following:

There are also limitations in the diversity of countries and professions represented in this study, as the majority of participants were from the UK (76%) and surgeons (83%). Findings may have been further developed with greater representation of non-UK settings, and other professions (e.g. anaesthetists, nurses, operating department practitioners). Moreover, the majority of surgeons who participated in this study were from developed countries. Risk mitigation in surgery may look very different in low- and middle-income countries with fewer resources, different healthcare infrastructures and more recent experience of epidemics. Capturing a wider range of international perspectives would inform the wider discourse on optimal practices for surgical procedures during the pandemic. (Page 19)

- You include a reference from Guba (28) but don't explain what it refers to explicitly - please explicitly mention in text if using reference.

Response: Thank you, we have now edited this section and removed the citation here.

- Have you any particular recommendations around communication and teamwork management as these variables played a significant role in to how teams adapted?

Response: This is an important consideration for this and future research. Whilst we did not feel we confidently offer specific 'recommendations' from this exploratory study, we do feel there is value in articulating our respondents' perspectives on what they found more or less helpful, as a foundation for future recommendation development. We have now added the following to the discussion to summarise lessons learned from the study:

We found that communication and teamwork were crucial to how teams adapted. Within this, we identified several practices that appeared helpful for reducing risks and optimising surgical practice during the duration of this pandemic (and potentially other adverse situations). Firstly, health care professionals articulated the importance of ensuring all aspects of the preoperative, operative and postoperative pathway was carefully mapped out to identify potential risks and staff roles. This enabled teams to minimise contact points and the number of people in the operating theatre to avoid potential infection transmission and contamination. It is important to note that pathway is likely to need to be reviewed frequently, in light of emerging evidence, experiences and concerns. To ensure practice is dynamic and cohesive, several participants in this study highlighted the value in ensuring all members of the surgical team were communicating regularly to discuss and reflect upon emerging evidence, guidance, experiences and concerns. Participants in our study also conveyed the importance of working together and developing local strategies to ensure PPE is donned and doffed safely and efficiently. This included visual aids (checklists,

signs/posters in theatre), having 'buddy systems' and engaging in team simulations.
(Pages 19)

Reporting Checklist:

- Make more explicit the objectives of the research (i.e. aims and objectives)

Response: Thank you, this has now been updated on page 5:

The study aimed to explore surgical teams' experiences of current practices during the pandemic. Specific objectives were to i) understand healthcare professionals' perceived risks of COVID-19 in surgery; and ii) to explore how surgical teams mitigated these risks. (Page 5)

'Qualitative approach' isn't clear from the methodology - would appear that it fell into the paradigm of constructivism. Was constructivist grounded theory

Response: We have now provided more information about this:

"This qualitative study adopted principles and techniques described by Glaser and Strauss (22) which enabled the inductive identification of codes from the data to generate findings that were grounded in the data (23) and the constant comparison approach, where new findings are systematically compared with existing data so that similarities and differences can be identified and emerging theories refined through the ongoing assimilation of data (22, 24). (Pages 5/6)

- 'Researcher characteristics' - I do think explaining more about who the researchers are and how they are situated within the environment they are researching is important. Do they know the participants? Were interviews done in person? in what setting?

Response: Details on interview setting added on page 6:

"Interviews were conducted either via University-approved secure video conferencing software or if not possible, telephone interviews were conducted instead."

As previously mentioned, researcher profiles have now been provided in Supplementary File 2.

- 'Setting; - were the settings public? one thing i noticed missing was the cancellation of many elective surgeries - did this feature within the findings? did individuals perceive this to be a 'risk' to their level of training/competency?

Response: Although the impact of the pandemic on training/competences was not identified as key finding, participants did discuss the implications of COVID-19 on surgical practice. We have now included more information about this on page 9:

All informants described how COVID-19 had had an impact on surgery. At the time of interviews, most elective procedures had been cancelled or postponed and only emergency procedures were being performed:

*HP30: We are doing absolute emergency. ... life and limb only...
Even to the extent that some appendicectomy operations are
being managed conservatively. So, they're just given antibiotics
and watch and wait. We know that a lot of people with
appendicitis ... if they're managed conservatively actually do fine.
And a lot of our appendixes now, ... are being managed
conservatively and watched. I think if you'd suggested that a year
ago, people would have said was insane.*

- 'Techniques to enhance trustworthiness' - this would be important to reference more. This article might be helpful - Irene Korstjens & Albine Moser (2018) Series: Practical guidance to qualitative research. Part 4: Trustworthiness and publishing, European Journal of General Practice

Response: Thank you for the suggestion. We have now included a section on 'strengths and limitations' on pages 18-19, which includes a reference to this paper and discusses the techniques that we adopted to enhance trustworthiness:

Strengths and limitations

Our study had some methodological limitations. Initially, individuals who were known to the study team were approached to consider participation and subsequent participants were identified via snowball sampling. It is therefore possible that the findings may not be representative of all healthcare professionals. The researchers in this study were known to have a particular interest in surgical innovation and responses may have also been influenced by participants' perceptions of what might constitute socially desirable answers, particularly given the emotive nature of some of the topics discussed in the interviews (e.g. resource constraint, balancing risks to patients versus risks to themselves). However, contrasting perspectives did emerge from our sample, and we sought to search for and report negative cases that did not align with recurring ideas. The findings have been reported with negative cases in mind, to present as comprehensive an account of key themes as possible (22).

It is important to note that interviews were conducted between March and May 2020, which has limitations for the transferability of results. Whilst we were able to capture initial responses as the pandemic unfolded in its first wave, future research is warranted to explore how teams continue to adapt during the later stages and aftermath of the pandemic. There are also limitations in the diversity of countries and professions represented in this study, as the majority of participants were from the UK (76%) and surgeons (83%). Findings may have been further developed with greater representation of non-UK settings, and other professions (e.g. anaesthetists, nurses, operating department practitioners). Moreover, the majority of surgeons who participated in this study were from developed countries. Risk mitigation in surgery

may look very different in low- and middle-income countries with fewer resources, different healthcare infrastructures and more recent experience of epidemics. Capturing a wider range of international perspectives would inform the wider discourse on optimal practices for surgical procedures during the pandemic. (Pages 18/19)

Well done again to the authors on this study. I look forward to seeing your responses.

Response: Thank you very much for your incredibly helpful and insightful comments.

Regards,

Dale F. Whelehan

Reviewer: 2

Dr. Molly Jarman, Brigham and Women's Hospital Biomedical Research Institute

Comments to the Author:

Thank you for the opportunity to review this manuscript reporting findings from an international qualitative study of surgical care during the first wave of the COVID-19 pandemic.

Introduction focuses on guidelines/practices in the UK, but study is international in scope - the intro would be strengthened with some mention of variation in guidelines across national systems.

Response: Thank you for the suggestion. We have now included examples of international guidelines, a reference to a systematic review of international guidelines (Moletta et al) and citations to demonstrate there is variation across these (in bold):

*“Guidelines were rapidly published (7) (8), including recommendations from The Royal Colleges (such as the Royal College of Surgeons of England, Royal College of Anaesthetists and The Intensive Care Society), Public Health England and the Department of Health, **the American College of Surgeons, and the World Health Organisation**. These have been produced based on expert opinion, surveys, consensus work and rapid reviews, and continue to evolve as evidence unfolds and **can include contradictory recommendations (9-12).**” (Page 5)*

Given variation in guidelines/policies across health systems, did interviewers modify questions/probes to address differences at sites outside the UK? For example, how PPE shortages in the US, or extraordinary stay at home orders in New Zealand?

Response: Thank you for raising this. The interviews were semi-structured so a topic guide ensured that the same key areas were covered across all interviews. This enabled participants to share their specific experiences of particular issues, such as PPE. To clarify this, we have updated the term 'discussion' with 'semi-structured interviews' to specify the consistency in interviewing by the different researchers (see page 6) and added the following text in bold:

Semi-structured interviews were directed by a topic guide to ensure that the same core areas were consistently covered amongst the team of interviewers, whilst allowing flexibility to pursue the detail that was salient to each participant (Supplementary File 3). Specific questions were open-ended to encourage the participant to talk about their own experiences of particular topics (“In your own words, can you talk me through how PPE has been used during COVID-19?”), experiences (“Can you describe a case where COVID-19 affected what happened in the operating theatre?”) and views (“In your opinion what are the key risks in surgery due to COVID-19?”). Interviewers were then able to incorporate the interviewee’s own terms and concepts into subsequent questions to follow up on specific issues raised and check the interviewer had understood correctly (27). At the end of the interview, participants were given an opportunity to raise relevant issues that had not already been covered (28). Regular team meetings allowed for the team to review the topic guide in light of findings and suggest potential changes” (Pages 6/7)

The findings of this study are valuable as we move through the second wave of the current pandemic, but perhaps more valuable in relation to future pandemic disease. The paper would benefit from some discussion of implications of "lessons learned" for future pandemic or other disasters. Similarly, how might these findings impact surgical care in non-pandemic/disaster settings?

Response: Thank you for the helpful suggestion. A section on lessons learned has now been added to the discussion:

We found that communication and teamwork were crucial to how teams adapted. Within this, we identified several practices that appeared helpful for reducing risks and optimising surgical practice during the duration of this pandemic (and potentially other adverse situations). Firstly, health care professionals articulated the importance of ensuring all aspects of the preoperative, operative and postoperative pathway was carefully mapped out to identify potential risks and staff roles. This enabled teams to minimise contact points and the number of people in the operating theatre to avoid potential infection transmission and contamination. It is important to note that pathway is likely to need to be reviewed frequently, in light of emerging evidence, experiences and concerns. To ensure practice is dynamic and cohesive, several participants in this study highlighted the value in ensuring all members of the surgical team were communicating regularly to discuss and reflect upon emerging evidence, guidance, experiences and concerns. Participants in our study also conveyed the

importance of working together and developing local strategies to ensure PPE is donned and doffed safely and efficiently. This included visual aids (checklists, signs/posters in theatre), having 'buddy systems' and engaging in team simulations. (Page 17)

Are any of the challenges discussed likely to persist after the pandemic subsides?

Response: Although it is not possible to hypothesise this from findings of the current study, we agree that this is an important point. We have now highlighted the need for further research about this in the discussion:

*Whilst we were able to capture initial responses as the pandemic unfolded in its first wave, **future research is warranted to explore how teams continue to adapt during the later stages and aftermath of the pandemic.** (Page 19)*

Reviewer: 3

Dr. Neal Kleiman, Houston Methodist DeBakey Heart and Vascular Center, Houston, Texas, USA

Comments to the Author:

The authors sampled a variety of individuals involved in the care of patients during the COVID era, predominantly in surgical specialties, and report their responses to an interview and discussion concerning the operational and emotional issues posed by the pandemic. Overall, the responses are fascinating to read. However, in terms of scientific study, there seems to be a tremendous amount of subjectivity here. The interviews may have been structured, but it's unclear how similar one interview was to another, whether the subjects were informed that the responses were going to be stratified into themes, and how the reported responses were selected. To this reviewer, the style is very reminiscent of a Ken Burns documentary (very popular on this side of the Atlantic) -- absolutely fascinating, extremely informative, often poignant -- but obviously subject to selection bias in terms of choosing exactly what to report. These caveats must be elucidated in the Discussion. The Methods section should accordingly describe to the best of the authors' ability, the criteria used to select the quotations that are included.

Response: Thank you for your positive comments, and we're pleased you enjoyed reading the paper. We have addressed each of your comments below.

In terms of scientific study, there seems to be a tremendous amount of subjectivity here.

Response: We set out to accurately present a comprehensive understanding of the stakeholders' realities and interpretations, and to conduct a systematic and inductive analysis that ensured all key findings were grounded in the data. To achieve this, we used the constant comparison technique (so that new findings were continually compared with the data). We have now clarified this in the manuscript:

Codes were identified inductively using methods derived from grounded theory methodology (20, 21) so that findings were grounded in the data to capture and

represent health care professionals' interpretations and experiences of surgical practice during a pandemic. Through the constant comparison technique, the codes were categorised into themes which were continually compared with the data. (Page 7)

Credibility is the equivalent of internal validity in quantitative research and is concerned with ensuring results are an accurate reflection of participants' experiences and perspectives (2). We employed several approaches to safeguard against the researchers' personal knowledge and experiences detracting from the participants' voices. For instance, in the method section, we have highlighted how we conducted various 'quality control' process to ensure we minimised personal perceptions in theme formation:

The coders met regularly to compare their coding and agree on the main broad categories encapsulating all codes. A subset of the transcripts, batched according to coder, were independently double coded by DE to determine broader consistency in coding approaches (26). Descriptive reports were written on key themes ('Guidance', 'Risk', 'In-theatre processes', 'Organisational', 'Service provision', 'Personal protective equipment (PPE)', 'Testing' and 'Physical and mental impact'). [...] producing descriptive reports provided an opportunity to demonstrate where there was overlap in key findings. Two coders (CO, MJ) also produced a summary on one theme, to explore any major differences in interpreting codes. As multiple qualitative researchers conducted the interviews, one researcher (DE) read all transcripts and synthesised the findings to ensure consistency in analysis and to enable strategic oversight of the findings(26). Results, with reference to the raw data, were discussed with members of the wider study team (including academic surgeons) and several research participants who were not part of the study team. (Page 8)

In the discussion, we have now added a section on measures taken to enhance the credibility of the results. We hope this clarifies the systematic, empirical approach to collecting and analysing data for this study.

Measures were undertaken to ensure rigour of the data in ensuring findings reflected the meaningful impact of COVID-19 on surgical teams. Rigour in qualitative research is determined by a range of criteria (31). Having multiple experienced qualitative methodologists conducting both the interviews, analysis and write-up (30) enabled analyst triangulation to strengthen credibility of the study (31, 37, 46). To ensure consistency in analysis one individual acted as an 'analytical auditor' and reviewed each transcript to enable strategic overview of the study findings and review the data for discrepancies or overstatements (30, 33, 46). Emerging themes, with reference to the raw data, were discussed with wider members of the study team (including academic surgeons), as well as research participants who were not part of the study

team, to ensure a good fit between researchers' interpretations and representation of the participants' experiences (30, 31, 47). (Pages 18/19)

It is unclear how similar one interview was to another:

Response: The interviews were semi-structured and conducted using a topic guide to ensure the same key areas were covered across all interviews. This enabled participants to share their specific experiences of particular issues, such as PPE. To clarify this, we have updated the term 'discussion' with 'semi-structured interviews' to specify the consistency in interviewing by the different researchers and added the following text in bold:

Semi-structured interviews were directed by a topic guide to ensure that the same core areas were consistently covered amongst the team of interviewers, whilst allowing flexibility to pursue the detail that was salient to each participant (Supplementary File 3). Specific questions were open-ended to encourage the participant to talk about their own experiences of particular topics ("In your own words, can you talk me through how PPE has been used during COVID-19?"), experiences ("Can you describe a case where COVID-19 affected what happened in the operating theatre?") and views ("In your opinion what are the key risks in surgery due to COVID-19?"). Interviewers were then able to incorporate the interviewee's own terms and concepts into subsequent questions to follow up on specific issues raised and check the interviewer had understood correctly (27). At the end of the interview, participants were given an opportunity to raise relevant issues that had not already been covered (28). Regular team meetings allowed for the team to review the topic guide in light of findings and suggest potential changes" (Pages 6/7)

It is unclear whether the subjects were informed that the responses were going to be stratified into themes.

Response: Participants were not explicitly informed about the technicalities of the analytical process as this is not generally an ethical requirement, but they were informed that they were participating in a research study, and were made aware of how their data would be used. This included explicit mention of the fact that their anonymised quotes may appear in publications. As we have mentioned, we did discuss the results with members of the study team (including academic surgeons), as well as research participants who were not part of the study team, to ensure a good fit between researchers' interpretations and representation of the participants' experiences and confirm check the credibility of the findings.

It is not clear how the reported responses were selected [...] The Methods section should accordingly describe to the best of the authors' ability, the criteria used to select the quotations that are included.

Response: Guidelines for qualitative research suggests that authors provide examples of the data to allow the reader to appraise the fit between the data and the author's understanding of them (3). We have now added the following to clarify this:

Results are supported with detailed quotations to support the interpretation of data (28, 37). Quotes that were deemed to best illustrate the themes were selected, with careful attention to showcasing different perspectives and negative cases, where relevant (Page 8)

References:

1. Braun V, Clarke V. Using thematic analysis in psychology. *Qualitative Research in Psychology*. 2006;3(2):77-101.
2. Korstjens I, Moser AJEJoGP. Series: Practical guidance to qualitative research. Part 4: Trustworthiness and publishing. 2018;24(1):120-4.
3. Elliott R, Fischer CT, Rennie DLJBjocp. Evolving guidelines for publication of qualitative research studies in psychology and related fields. 1999;38(3):215-29.